# Comparative Study of Acute Stress in Infants Undergoing Percutaneous Achilles Tenotomy for Clubfoot vs. Peripheral Line Placement

**DOI:** 10.3390/children11060633

**Published:** 2024-05-24

**Authors:** Anna Ey Batlle, Iolanda Jordan, Paula Miguez Gonzalez, Marta Vinyals Rodriguez

**Affiliations:** 1Equipo Ponseti Dra. Anna Ey, Clínica Diagonal, 08950 Barcelona, Spain; 2Hospital Sant Joan de Déu, 08950 Barcelona, Spain; 3Faculty of Medicine and Health Sciences, University of Barcelona, 08907 Barcelona, Spain

**Keywords:** percutaneous Achilles tenotomy, local anesthesia, Ponseti method, pain, infant

## Abstract

Introduction: Percutaneous tenotomy of the Achilles tendon is a procedure that is part of the Ponseti method for clubfoot correction. The need to apply general anesthesia or sedation for this procedure is controversial. The objective of this study is to compare the acute stress generated in infants by percutaneous Achilles tenotomy under local anesthesia vs. peripheral line placement. Material and methods: This cross-sectional study compares the discomfort experienced by 85 infants undergoing percutaneous Achilles tenotomy with local anesthesia with that experienced by 39 infants undergoing peripheral line placement. The following parameters were determined: the duration of the procedure, crying time, average crying intensity, and maximum crying intensity. Other data recorded included the infant’s age and complications arising during the procedure. Results: The mean ages of these patients were 1.95 and 2.18 months, respectively. The following data were obtained: the mean duration of the procedure for Group A was 8.13 s and for Group B it was 127.43 s; the mean duration of crying for Group A was 84.24 s and for Group B it was 195.82 s; the mean intensity of crying for Group A was 88.99 dB and for Group B it was 100.98 dB; and the maximum crying intensity for Group A was 96.56 dB and for Group B it was 107.76 dB. Conclusions: Percutaneous Achilles tenotomy can be safely performed as an outpatient procedure, under local anesthesia. This method generates less discomfort than peripheral line placement.

## 1. Introduction

Clubfoot is the most common congenital deformity affecting the foot and leg. Diagnosis often occurs during the 20-week morphology ultrasound scan during pregnancy. The etiology is idiopathic. The incidence is 1 in every 1000 newborns. The Ponseti method is the gold standard treatment for clubfoot and involves a gradual correction utilizing serial manipulation and casting to gradually correct the deformity. In a substantial majority of cases, approximately 95%, a percutaneous Achilles tenotomy is conducted [1,2,3,4,5]. Initially introduced by Ponseti in 1996, this method entails the sequential application of plaster casts from the toes to the groin alongside a percutaneous Achilles tenotomy, which boasted an impressive execution rate of 98.9%. Remarkably, this technique not only eradicated residual pain but also achieved complete rectification of the deformity’s components [2].

For optimal results, the percutaneous Achilles tenotomy should coincide with the attainment of 60–70° of abduction while maintaining the heel in a valgus position. Failure to execute the tenotomy under these conditions, leading to forced dorsiflexion within the casts, could precipitate deformities like the “rocker foot” or talus flattening [2].

Despite its minimally invasive nature and minimal documented complications [6,7], this procedure still instigates debates regarding the necessity of performing it under general anesthesia or sedation in infants. Ponseti’s original work and subsequent publications advocate for conducting the tenotomy under local anesthesia in an outpatient setting [2,8,9]. This preference underscores the method’s simplicity, requiring only basic surgical instruments. However, achieving consensus on the ideal approach remains elusive. Moreover, follow-up care is crucial, involving the use of foot abduction braces for up to four years to prevent relapse. Research also suggests that parent education and adherence to brace use play critical roles in the long-term success of the treatment. Additionally, ongoing assessments throughout the child’s growth are essential to address any adjustments in treatment and monitor for potential complications. This comprehensive approach highlights the Ponseti method’s effectiveness, balancing between non-invasive techniques and rigorous post-treatment care, which has made it a preferred choice in pediatric orthopedics worldwide.

Vinyals et al. [8] examined the procedure and quantified the stress produced when infants underwent percutaneous Achilles tenotomy under local anesthesia in the outpatient clinic. This study concluded that the procedure is safe, rapid, and produces minimal, tolerable discomfort for the patient for around 90 s. The caregivers consulted were very satisfied with the safety of the procedure and no complications were reported.

To our knowledge, no previous study has considered the question of the optimum approach to preventing distress during percutaneous Achilles tenotomy, although some have analyzed the protocol applied during the procedure [10,11] making reference to possible suffering when it is performed with local anesthesia, as a justification for applying sedation or general anesthesia. In view of this consideration, we decided to compare the results obtained from percutaneous Achilles tenotomy in infants with congenital clubfoot with those obtained from the placement of a peripheral line as part of the protocol for a preoperative blood test to perform the same surgical procedure in infants using general anesthesia. Peripheral line placement is a globally accepted process that is usually carried out without any type of anesthesia.

Our comparative analysis aims to provide a more nuanced understanding of the pain management strategies during such pediatric procedures. Furthermore, we include a review of the psychological impact on infants and their parents, investigating whether the short duration of pain associated with the tenotomy could justify avoiding the risks associated with general anesthesia. This study expands on the existing literature by exploring not just the procedural efficacy but also patient and caregiver experiences, which are crucial for holistic patient care. We also consider the long-term effects of repeated anesthesia exposure in infants, aligning with recent research emphasizing the need for minimal anesthesia use in early childhood. By gathering comprehensive data, this research could potentially reshape guidelines and improve practice in pediatric orthopedics.

In analyzing the acute stress that may be generated in percutaneous Achilles tenotomy as an outpatient procedure for infants, performed under local anesthesia, vs. peripheral line placement, we must quantify the suffering produced and, moreover, determine the impact made on related variables. This includes evaluating both the immediate and short-term physiological responses such as heart rate and cortisol levels, as well as the emotional responses of both the infant and the caregivers. By comprehensively assessing these factors, we aim to better understand the broader implications of pain management techniques in pediatric procedures. This research will contribute to optimizing patient comfort and could potentially lead to revised protocols that prioritize minimal distress while maintaining procedural efficacy.

## 2. Materials and Methods

A cross-sectional comparison study was made of 85 patients with clubfoot who underwent percutaneous tenotomy of the Achilles tendon after corrective casting, in accordance with the Ponseti method [2], and of 39 patients who underwent peripheral line placement prior to general anesthesia.

The inclusion criteria were that the infants should be infants aged 0 to 24 months and either scheduled to undergo a percutaneous Achilles tenotomy to correct congenital clubfoot, following the Ponseti method, or be scheduled to undergo peripheral line placement prior to general anesthesia.

Any patients with neurological involvement or in whom an associated pathology was detected were excluded from the study.

In every case, the percutaneous Achilles tenotomy was performed as an outpatient procedure by the team led by Dr. Anna Ey at the Diagonal Clinic in Barcelona, Spain, and the peripheral line placement took place at Sant Joan de Déu Hospital (Barcelona, Spain) between 1 January and 31 December 2019. The tenotomies were always performed by an orthopedic surgeon with 27 years of experience in clubfoot and assisted by a podiatrist with 9 years of experience.

In the study, a comprehensive analysis was conducted on various variables relevant to assess the efficacy and safety of percutaneous Achilles tendon tenotomy under local anesthesia in pediatric patients. The analyzed variables included patients’ age, mean procedure time, mean crying time, mean crying intensity, and maximum crying intensity. Additionally, the presence or absence of complications related to tenotomy or non-cannulation of the peripheral line, as well as the results of the caregiver satisfaction survey, were considered.

A description of the surgical procedure for percutaneous Achilles tenotomy under local anesthesia (Figure 1) is given as follows:Incision Site Preparation: The procedure begins by applying topical local anesthesia to the Achilles tendon area and covering it with an occlusive dressing to ensure proper absorption and effectiveness of the anesthetic.Asepsis and Surgical Field Preparation: Meticulous asepsis of the surgical area is performed to minimize the risk of infection during the procedure.Incision Making: A medial to lateral incision is made approximately 1 cm proximal to the calcaneal insertion of the Achilles tendon using a cataract knife or other suitable surgical instrument.Complete Tendon Sectioning: With careful precision, a complete sectioning of the Achilles tendon is performed, ensuring adequate separation of tendon fibers and avoiding damage to surrounding structures. Upon completion of the sectioning, localized pressure is applied to the area to control any bleeding and promote hemostasis.Incision Closure: Skin closure strips are used to securely close the incision and provide optimal wound healing. This method of closure minimizes the possibility of suture-related complications and promotes postoperative recovery.Postoperative Immobilization: A corrective cast is applied to maintain dorsiflexion of the foot at an angle of 15–20 degrees and the hyperabduction achieved with previous casts at an angle of 60–70 degrees for a period of 14 days. This immobilization helps maintain proper foot alignment and facilitates optimal healing of the Achilles tendon.

This comprehensive approach to the surgical procedure ensures optimal patient care and satisfactory outcomes in terms of therapeutic efficacy and safety. Additionally, detailed analyses of the mentioned variables provide a comprehensive understanding of the study results and support informed clinical decision making in medical practice.
Figure 1Percutaneous Achilles tenotomy under local anesthesia. The tenotomy is performed 1 cm proximal to the insertion of the calcaneus or 1 cm proximal to the posterior fold present in feet with a lot of equinus. The confirmation of complete sectioning is performed using ultrasound.
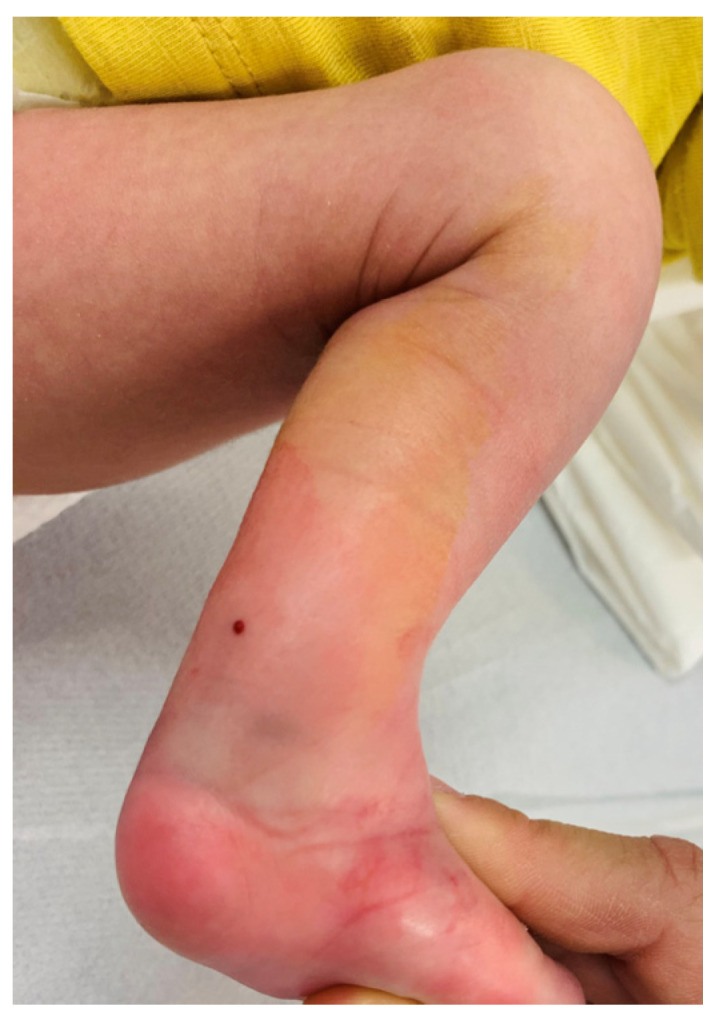


Measurement of crying time and intensity:Utilizing the Voice Memos application on the iPhone X, an audio recording is conducted, positioning the smartphone approximately one meter away from the infant.The commencement of the recording coincides with the initiation of the surgical procedure, marked by the insertion of the scalpel, and concludes upon the culmination of the surgical procedure, signified audibly by a distinct click indicating the achievement of tenotomy and subsequent descent of the calcaneus.The intensity of the infant’s crying is quantified through the employment of the Decibel 10 application, specifically designed for the iPhone X platform.The latter recording period commences three minutes prior to the initiation of the surgical procedure and extends until the cessation of the infant’s crying, thereby encompassing the entirety of the vocal response.

### 2.1. Ethical Approval

In every case, utmost care was taken to ensure the ethical integrity of the study. Prior to any involvement, the infant’s parent or legal guardian was provided with comprehensive information regarding the study’s objectives, procedures, and potential risks, and they willingly provided their informed consent by signing a statement. All actions undertaken within the study adhered strictly to the ethical standards outlined by the institution. Furthermore, the experimental protocol underwent rigorous review and received approval from a designated representative of the CEIm Fundacio Sant Joan De Deu (PIC-103-18). Throughout the entire duration of the study, strict adherence to the principles set forth in the 1964 Helsinki Declaration of Ethical Principles was maintained, ensuring the protection and well-being of all participants.

### 2.2. Statistical Analysis

The data collected for this study underwent meticulous statistical analysis to derive meaningful insights. Analysis was conducted utilizing both an Excel spreadsheet and the SPSS 22.0 statistical program developed by IBM Corp. (Armonk, NY, USA). Prior to analysis, the normality of the data distribution was assessed using the Kolmogorov–Smirnov test, while homogeneity of variance was evaluated using the Levene test. Following these assessments, a comprehensive descriptive analysis of the variables was undertaken to characterize the dataset. Bivariate analysis ensued, with normally distributed data being subjected to the Student *t*-test, while non-normally distributed data underwent analysis via the Mann–Whitney U and Kruskal–Wallis non-parametric tests. The significance threshold was set at *p* < 0.05 for all analyses, and all tests were conducted as two-sided evaluations to ensure robustness and accuracy in the interpretation of results.

## 3. Results

The study data presented a comprehensive analysis of 85 patients diagnosed with clubfoot who underwent percutaneous tenotomy of the Achilles tendon, constituting Group A, alongside 39 patients who underwent peripheral line placement as a precursor to surgery under general anesthesia, forming Group B.

In Group A, the cohort’s average age was 1.95 months (±1.63), with a range spanning from birth to 7 months. Conversely, Group B exhibited a slightly higher average age of 2.18 months (±1.12), with the age range extending from birth to 8 months (as detailed in Table 1). This demographic distribution establishes a comparative baseline between the two groups, crucial for subsequent analyses.

The procedural duration analysis elucidated stark differences between the two cohorts. Group A displayed a notably brief average procedural time of 8.13 s (SD 5.97), whereas Group B necessitated significantly longer at an average of 127.43 s (SD 130.74). This substantial contrast proved statistically significant, with a *p*-value < 0.001, emphasizing the efficiency of percutaneous tenotomy over peripheral line placement (refer to Table 1 for detailed statistical summaries).

Further delving into patient experiences, the examination of crying time revealed compelling insights. Group A exhibited a substantially shorter mean crying duration, averaging at 84.24 s (SD 46.73), compared to Group B’s markedly prolonged mean of 195.82 s (SD 141.29). Once again, statistical analysis underscored the significance of this contrast, with a *p*-value < 0.001, highlighting the potential discomfort differentials experienced by patients in the respective groups during the pre-surgical phase.

Moreover, the assessment of crying intensity unveiled compelling disparities between the cohorts. Group A showcased a notably lower mean crying intensity, registering an average of 88.99 dB (SD 8.61), in contrast to Group B’s elevated mean of 100.98 dB (SD 9.76). This substantial difference in intensity levels, validated by a *p*-value < 0.001, accentuates the potential variations in pain perception and tolerance between the groups, further emphasizing the potential benefits of percutaneous tenotomy in mitigating patient distress.

Additionally, scrutinizing the maximum crying intensity elucidated notable contrasts between the cohorts. Group A maintained a lower peak intensity, averaging at 96.56 dB (SD 8.19), while Group B exhibited a notably higher peak at 107.76 dB (SD 10.30). This statistical significance (*p*-value < 0.001) underscores the potential variations in pain response and management efficacy between the two procedural approaches.

Subsequent null hypothesis testing, encompassing the mean procedural time, crying duration, mean crying intensity, and maximum crying intensity, yielded compelling results. The resultant *p*-value of <0.001 unequivocally rejected the null hypothesis, affirming the substantive differences between Group A and Group B. This validation underscores the superior outcomes associated with percutaneous tenotomy of the Achilles tendon in terms of procedural efficiency and patient comfort, compared to peripheral line placement.

In conclusion, these findings elucidate the tangible advantages of percutaneous tenotomy over peripheral line placement as a pre-surgical procedure for clubfoot correction. Not only does it streamline procedural efficiency, but it also minimizes patient distress, underscoring its pivotal role in optimizing clinical outcomes and enhancing patient experiences in the management of clubfoot. Such empirical insights are invaluable for informing evidence-based clinical practices and optimizing patient care pathways in orthopedic surgery.

None of the patients experienced any complications due to bleeding or the application of local anesthesia.

After the tenotomy, the parents were asked to complete a 5-item survey regarding their satisfaction with the procedure; 96% of respondents indicated they felt completely satisfied in this respect. This statistic underscores the positive reception and endorsement of the procedure by the parents of the patients. The overwhelming majority expressing complete satisfaction highlights the successful outcomes and the fulfillment of expectations regarding the tenotomy. Such high levels of satisfaction not only reflect positively on the procedure itself but also on the healthcare professionals involved, affirming their expertise and ability to deliver satisfactory outcomes. Overall, these findings validate the tenotomy as a well-tolerated and satisfactory intervention in the treatment of the respective condition, providing reassurance to both patients and their families.

## 4. Discussion

The aim of this study is to compare two surgical procedures, percutaneous Achilles tenotomy vs. peripheral line placement, in terms of the discomfort produced when conducted in infants on an outpatient basis.

To our knowledge, no previous research in this respect has been conducted. The reason for doing so, in view of the results obtained in a preliminary investigation [8], is that peripheral line placement is widely acknowledged to be feasible without sedation; moreover, it is necessary for any preoperative examination and provides a good basis for comparison with tenotomy because both interventions may provoke acute stress in the infant.

A long-standing gap in quantifying the extent of infant distress during percutaneous tenotomy with local anesthesia, conducted in the outpatient clinic, together with healthcare professionals’ subjective evaluation of this distress, has led to parents being given incorrect information in this respect. The main aim of the present study is to clarify this question.

Although various scales of infant stress have been proposed, such as the Premature Infant Pain Profile [12] and the Neonatal Infant Pain Scale [13], all of these focus on the moment at which discomfort increases in the hospital environment or on determining certain behavioral and physiological parameters, for example, during admission to intensive care. However, none consider the acute distress that may be provoked in infants. Accordingly, the present study addresses this question in terms of the infant’s crying time and intensity. In the latter respect, we observed a mean negative difference of 12 dB between the infants who underwent percutaneous tenotomy, on the one hand, and peripheral line placement, on the other. In other words, the first of these interventions provokes less stress and discomfort in the infant.

When examining the effects of percutaneous Achilles tenotomy on infants, our primary focus lies on understanding the duration and intensity of crying. This is crucial as the procedure is recognized to induce stress in infants. The core inquiry revolves around comprehending the extent and duration of distress experienced by infants during treatment in outpatient settings, particularly when accompanied by their parent(s). In such settings, the immediate measurement of other physiological variables such as oxygen saturation, heart rate, or respiratory rate may not be readily accessible. Therefore, by homing in on the duration and intensity of crying, we aim to grasp a tangible indicator of the infant’s discomfort and distress throughout the procedure. This approach enables us to evaluate the overall impact of percutaneous Achilles tenotomy on the infant’s well-being in the absence of real-time monitoring of other physiological parameters.

In several protocols, it is proposed that percutaneous Achilles tenotomy be performed with sedation or general anesthesia [10,11,14]. According to Herzenberg and Parada, this approach is necessary in order to avoid suffering, to better control the patient’s condition, and to increase the safety of the procedure by avoiding injury to neighboring anatomical structures. In our study, however, there were no problems regarding control of the patient’s limb that might have jeopardized patient safety. An important consideration is that the infants were at all times accompanied by their parents, which produces a calming effect. This consideration is corroborated by Rangasamy et al. [15], for whom the best methods of non-medicated sedation include the presence of the parents, breastfeeding or formula feeding, and skin-to-skin contact. However, the latter study did not include a comparative analysis of the benefit produced.

In our study, there were no complications, such as the pseudoaneurysm of the posterior tibial artery that has been described by Dobbs [7]. In previous work [6,7], researchers analysed the case of tenotomy performed under local anesthesia in terms of bleeding and possible complications but did not consider the discomfort caused to the infant. Thus, no comparison was made between different short, painful procedures, such as tenotomy, taking the duration and intensity of crying as an indication of the pain experienced [11]. In the present study, it was observed that the procedure with a local anesthetic, compared to peripheral line placement, greatly reduces the time required to perform the procedure; in fact, it is 16 times faster.

With respect to the use of anesthetic drugs, the choice between local or general anesthesia for procedures like tenotomy in infants is a subject of ongoing debate. Some studies suggest that administering general anesthesia to infants under three months old carries a heightened risk of complications, such as neonatal apnea [16,17]. Given these risks, it is crucial to investigate whether sedation or general anesthesia is truly necessary for relatively minor procedures. Moreover, the use of general anesthesia (but not local anesthesia) in infants has been associated with potential long-term issues related to the anesthetic drugs, such as attention deficits during childhood [18,19]. According to researchers like Dimaggio, Wilder, and Vutskits, infants exposed to general anesthesia may also experience sensitive and somatosensory changes in response to painful stimuli, neurosensory development problems, alterations in emotional behavior, and impaired learning capabilities [18,19,20].

This growing body of evidence points to the need for a careful reassessment of anesthesia protocols in pediatric surgery, particularly for non-critical procedures. The potential for lasting neurodevelopmental impact should lead to more conservative use of general anesthesia in early childhood, advocating for the use of local anesthesia where feasible. Additionally, the medical community must consider developing and adopting new techniques and technologies that minimize the need for invasive anesthesia methods, thus safeguarding the neurodevelopment of the youngest patients while effectively managing pain. Further research into the effects of different anesthesia types on infant health outcomes is essential to ensure that clinical practices adapt to these findings, prioritizing the long-term well-being of pediatric patients.

Although, in accordance with the Ponseti method, a percutaneous tenotomy procedure should be carried out in the operating room [21], it can be performed in an outpatient clinic and with local anesthesia. This offers the following advantages: the family can be present, there is a lower financial cost, and there is no need for preoperative fasting. The use of general anesthesia, on the other hand, requires a preoperative study (including a preoperative analysis and the placement of a peripheral line), the separation of the infant from its caregivers, and a six-hour period of fasting that is usually associated with a long period of crying.

A systematic review by Rangasamy et al. [15] concluded that percutaneous tenotomy under local anesthesia, as an outpatient procedure, was both safe and preferable to general anesthesia, which they consider unnecessary, as there are no significant differences in the outcomes produced, while outpatient care has a much lower financial cost and is not associated with long-term adverse effects.

The present study has some limitations. Firstly, it assumes that infant crying correlates directly with the experience of acute pain, yet this assumption lacks validation within the context of this study or broader research. This gap highlights the need for further investigation into the relationship between crying behavior and pain sensation in infants, potentially through additional studies or experimental validation.

Moreover, this study does not account for the possible influence of external factors such as the infant’s general health, environment during the procedure, or the presence of the caregiver, which may affect the crying response. The complexity of infant responses to pain could also be influenced by developmental stages, making it challenging to generalize findings across different age groups without a stratified analysis. Additionally, the reliance on observable behaviors such as crying may overlook subtler physiological responses to pain that are not as easily detectable without the use of specialized monitoring equipment. To enhance the robustness of future research, incorporating a multi-dimensional approach to measure physiological indicators, such as heart rate variability and stress hormones, alongside behavioral observations would provide a more comprehensive understanding of infants’ pain experiences.

Additionally, the study recognizes the rarity of congenital clubfoot, with an incidence rate of only 1–2 per 1000 live newborns. This low prevalence may impact the generalizability and significance of the study’s findings, potentially limiting the broader applicability of the results. Despite this challenge, the study maintains confidence in the validity of its statistical analysis, emphasizing the importance of a rigorous methodology despite inherent limitations in the sample size due to the nature of the condition under study.

Despite these limitations, the study’s data serve a valuable purpose in providing informative and scientifically grounded insights for families of patients with clubfoot. By offering detailed information about the procedure and its implications, the study aims to empower families to make well-informed decisions, thereby fostering a sense of reassurance and facilitating the decision-making process. This contribution to patient education and support underscores the practical relevance and impact of the study’s findings within clinical practice, despite the acknowledged constraints.

The limitations of this study include equating crying to acute infant pain, which has not been validated or used in other studies, and the low incidence of congenital clubfoot. Due to the rarity of congenital clubfoot (1/1000 live births), we consider the sample size to be more than sufficient, although the pathology limits the sample size. The authors conclude that subjecting children to general anesthesia is unnecessary since there are no significant differences in outcomes between performing the procedure on an outpatient basis versus in the operating room, the cost is much lower if performed in the clinic, and the procedure is not associated with long-term adverse effects.

## 5. Conclusions

Percutaneous Achilles tenotomy, a procedure commonly conducted under local anesthesia on an outpatient basis, proves to be notably less discomforting compared to the alternative method of peripheral line placement. This conclusion is supported by empirical evidence showcasing significantly reduced crying times during the procedure, with 84.24 s recorded for Achilles tenotomy compared to 195.82 s for peripheral line placement. Moreover, the intensity of crying, measured in decibels, further underscores the diminished discomfort associated with Achilles tenotomy, with levels reaching 88.99 dB compared to 100.98 dB for peripheral line placement. These findings underscore the efficacy and patient comfort provided by percutaneous Achilles tenotomy, offering a less distressing option for outpatient procedures.

## Figures and Tables

**Table 1 children-11-00633-t001:** Relationship between measures of variables of Groups A and B.

	Group A (*n* = 85)			Group B (*n* = 39)			
	Mean	SD	Min/Max	Mean	SD	Min/Max	*p*-Value
Age (months)	1.95	1.63	0/7	2.18	1.12	0/8	<0.001
Procedure time (seconds)	8.13	5.97	2.1/33.5	127.43	130.74	14.7/558.2	<0.001
Mean crying time (seconds)	84.24	46.73	13.7/267.4	195.82	141.29	80.6/670.3	<0.001
Mean crying intensity (dB)	88.99	8.61	67.6/107.3	100.98	9.76	86.3/139.6	<0.001
Maximum crying intensity (dB)	96.56	8.19	79.3/116.2	107.76	10.30	93.6/142.8	<0.001

## Data Availability

Data are contained within the article.

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
