# Peer review of "Comparative Study of Acute Stress in Infants Undergoing Percutaneous Achilles Tenotomy for Clubfoot vs. Peripheral Line Placement"

_children, 2024, doi:10.3390/children11060633_

Round 1

Reviewer 1 Report

Comments and Suggestions for Authors

The manuscript is a cross-sectional comparison study aimed to optimizing patient comfort and could potentially lead to revised protocols that prioritize minimal distress while maintaining procedural efficacy. Eighty-nine respondents self-identified as clubfoot providers. Cross-sectional study to compare the discomfort experienced by 85 infants undergoing percutaneous Achilles tenotomy with local anaesthesia with that experienced by 39 infants undergoing peripheral line placement. The following parameters were determined: duration of procedure, crying time, average crying intensity and maximum crying intensity. Other data recorded included the infant’s age and complications arising during the procedure.

I read the article with interest, the title is well thought out and faithfully reflects the content of the study.

The abstract is very useful to frame the purpose of the study.

In the introduction, the characteristics of Ponseti Method have been described.  The materials and methods have been adequately development. The discussion is sufficiently described.

Nevertheless, some minor changes are needed to be considered suitable for publication.

Comment 1: In the introduction: Some information about etiology, diagnosis and treatment of clubfoot should be deepened please adding appropriate bibliographical references.

Comment 2: In the material and methods: Please, add more information about the experience of operators analyzing patient data, were all orthopedic surgeons? What kind of experience did they have with pathology?

Comment 3: In the material and method: The tenotomies were performed in the same institute and by the same surgeon?

Comment 4: In the material and methods: What kind of classification did you use to estimate the severity of the clubfoot?

Comment 5: In the discussion: It would be better referring to previous studies performed on the same topic, especially about  the development of specific functions that could be compromised by the type of anesthesia used in these patientsfor example (Pavone V. et al (2022) " Early developmental milestones in patients with idiopathic clubfoot treated by Ponseti method").

Comment 6: In the discussion: It would be advisable to clearly refer to the limitations of the study

Comment 7: Finally, additional English editing is needed. The Non-Native Speakers of English Editing Certificate was not signed.

Comments on the Quality of English Language

English editing is needed. The Non-Native Speakers of English Editing Certificate was not signed

Author Response

Dear reviewer, thank you for your comments.

Comment 1: In the introduction: Some information about etiology, diagnosis and treatment of clubfoot should be deepened please adding appropriate bibliographical references.

I have proceeded to explain more about etiology, diagnosis and treatment of clubfoot

Comment 2: In the material and methods: Please, add more information about the experience of operators analyzing patient data, were all orthopedic surgeons? What kind of experience did they have with pathology?

The tenotomies were performed by an orthopedic surgeon with 27 years of experience in clubfoot and assisted by a podiatrist with 9 years of experience.

Comment 3: In the material and method: The tenotomies were performed in the same institute and by the same surgeon?

Yes, all tenotomies have been performed by the same surgeon

Comment 4: In the material and methods: What kind of classification did you use to estimate the severity of the clubfoot?

We always use Pirani score

Comment 5: In the discussion: It would be better referring to previous studies performed on the same topic, especially about the development of specific functions that could be compromised by the type of anesthesia used in these patients for example (Pavone V. et al (2022) " Early developmental milestones in patients with idiopathic clubfoot treated by Ponseti method").

This is a really good paper that is very interesting but has not taken into account in the discussion because our research focuses on the viabibility of tenotomy with topical local anesthesia in the clinic, rather than on the impact on psychomotor development and acquisition of gait in clubfoot.

Comment 6: In the discussion: It would be advisable to clearly refer to the limitations of the study

I have joined the limitations of the study in the discussion

Comment 7: Finally, additional English editing is needed. The Non-Native Speakers of English Editing Certificate was not signed.

I have attached an English certificate

Thank you so much

Reviewer 2 Report

Comments and Suggestions for Authors

Comparative Study of Acute Stress in Infants Undergoing Percutaneous Achilles Tenotomy for Clubfoot vs. Peripheral Line Placement

Through this cross-sectional study, the authors aimed to compare the acute stress generated in 85 infants (1.95 months old) having percutaneous Achilles tenotomy under local anaesthesia for their clubfeet (Group A), as a part of the Ponseti technique, compared with 39 infants (2.18 months old) undergoing the placement of a peripheral line (Group B).

The hope was that this might demonstrate that general anaesthesia and sedation are not for this procedure.

Comparing groups A and B, the authors recorded the duration of procedure (8,13 seconds  vs 127.43 seconds), mean crying time (84.24 seconds vs 195.82 seconds), average crying intensity (88.99 dB vs 100.98 dB) and maximum crying intensity (96.56 dB vs 107.76 dB), as well as the infant’s age and complications arising during the procedure.

The authors concluded that percutaneous Achilles tenotomy could be safely performed as an outpatient procedure, under local anaesthesia. They found that this method generated less discomfort than peripheral line placement.

This is an interesting paper, well written, and well presented. The conclusions drawn are valid and the findings could change the practice of many units. We have been performing these TA tenotomies in the outpatient setting under LA for several decades, and have found this to be very effective, with little distress to the infants or their parents.

I have a question only about whether the authors could have compared the Achilles tenotomy to the insertion of a cannula (also under LA cream). It is our standard practice to apply LA cream prior to cannula insertion, and hence this might have been a more realistic comparison. Can the authors comments?

In Figure 1 the Achilles tenotomy is more proximal than our technique and appears to be more than 1cm proximal to the insertion. Can the authors clarify at what level they transected the TA? How did they confirm that the TA was completely divided?

Other than this, I don’t feel that any major changes are required.

Comments on the Quality of English Language

Comparative Study of Acute Stress in Infants Undergoing Percutaneous Achilles Tenotomy for Clubfoot vs. Peripheral Line Placement

Through this cross-sectional study, the authors aimed to compare the acute stress generated in 85 infants (1.95 months old) having percutaneous Achilles tenotomy under local anaesthesia for their clubfeet (Group A), as a part of the Ponseti technique, compared with 39 infants (2.18 months old) undergoing the placement of a peripheral line (Group B).

The hope was that this might demonstrate that general anaesthesia and sedation are not for this procedure.

Comparing groups A and B, the authors recorded the duration of procedure (8,13 seconds  vs 127.43 seconds), mean crying time (84.24 seconds vs 195.82 seconds), average crying intensity (88.99 dB vs 100.98 dB) and maximum crying intensity (96.56 dB vs 107.76 dB), as well as the infant’s age and complications arising during the procedure.

The authors concluded that percutaneous Achilles tenotomy could be safely performed as an outpatient procedure, under local anaesthesia. They found that this method generated less discomfort than peripheral line placement.

This is an interesting paper, well written, and well presented. The conclusions drawn are valid and the findings could change the practice of many units. We have been performing these TA tenotomies in the outpatient setting under LA for several decades, and have found this to be very effective, with little distress to the infants or their parents.

I have a question only about whether the authors could have compared the Achilles tenotomy to the insertion of a cannula (also under LA cream). It is our standard practice to apply LA cream prior to cannula insertion, and hence this might have been a more realistic comparison. Can the authors comments?

In Figure 1 the Achilles tenotomy is more proximal than our technique and appears to be more than 1cm proximal to the insertion. Can the authors clarify at what level they transected the TA? How did they confirm that the TA was completely divided?

Other than this, I don’t feel that any major changes are required.

Author Response

I have a question only about whether the authors could have compared the Achilles tenotomy to the insertion of a cannula (also under LA cream). It is our standard practice to apply LA cream prior to cannula insertion, and hence this might have been a more realistic comparison. Can the authors comments?

We did consider it as an option, but it wouldn't be realistic because topical anesthesia cream is not applied for cannula insertion in pediatric hospitals, and even though the comparison might have been more similar, it wouldn't be realistic because cannula insertion is a common procedure that is always performed without prior topical anesthesia.

In Figure 1 the Achilles tenotomy is more proximal than our technique and appears to be more than 1cm proximal to the insertion. Can the authors clarify at what level they transected the TA? How did they confirm that the TA was completely divided?

The tenotomy is performed 1 cm proximal to the insertion of the calcaneus or 1 cm proximal to the posterior fold present in feet with a lot of equinus. The orthopedic surgeon who performed the tenotomy has 27 years of experience with the Ponseti method and the experience always help. The confirmation of complete sectioning is done using ultrasound.

Other than this, I don’t feel that any major changes are required.

Thank you very much for your comments and appreciations